# The Essential Role of H_2_S-ABA Crosstalk in Maize Thermotolerance through the ROS-Scavenging System

**DOI:** 10.3390/ijms241512264

**Published:** 2023-07-31

**Authors:** Jia-Qi Wang, Ru-Hua Xiang, Zhong-Guang Li

**Affiliations:** 1School of Life Sciences, Yunnan Normal University, Kunming 650092, China; jiaqi.wang@sansure.com.cn (J.-Q.W.);; 2Engineering Research Center of Sustainable Development and Utilization of Biomass Energy, Ministry of Education, Kunming 650092, China; 3Key Laboratory of Biomass Energy and Environmental Biotechnology, Yunnan Province, Yunnan Normal University, Kunming 650092, China

**Keywords:** abscisic acid, hydrogen sulfide, maize seedling, ROS-scavenging system, thermotolerance

## Abstract

Hydrogen sulfide (H_2_S) and abscisic acid (ABA), as a signaling molecule and stress hormone, their crosstalk-induced thermotolerance in maize seedlings and its underlying mechanism were elusive. In this paper, H_2_S and ABA crosstalk as well as the underlying mechanism of crosstalk-induced thermotolerance in maize seedlings were investigated. The data show that endogenous levels of H_2_S and ABA in maize seedlings could be mutually induced by regulating their metabolic enzyme activity and gene expression under non-heat stress (non-HS) and HS conditions. Furthermore, H_2_S and ABA alone or in combination significantly increase thermotolerance in maize seedlings by improving the survival rate (SR) and mitigating biomembrane damage. Similarly, the activity of the reactive oxygen species (ROS)-scavenging system, including enzymatic antioxidants catalase (CAT), ascorbate peroxidase (APX), guaiacol peroxidase (POD), glutathione reductase (GR), monodehydroascorbate reductase (MDHAR), dehydroascorbate reductase (DHAR), and superoxide dismutase (SOD), as well as the non-enzymatic antioxidants reduced ascorbic acid (AsA), carotenoids (CAR), flavone (FLA), and total phenols (TP), was enhanced by H_2_S and ABA alone or in combination in maize seedlings. Conversely, the ROS level (mainly hydrogen peroxide and superoxide radical) was weakened by H_2_S and ABA alone or in combination in maize seedlings under non-HS and HS conditions. These data imply that the ROS-scavenging system played an essential role in H_2_S-ABA crosstalk-induced thermotolerance in maize seedlings.

## 1. Introduction

Hydrogen sulfide (H_2_S), which is similar to other signaling molecules, has an inversely physiological effect at low (as a signaling molecule) and high (cytotoxic agent) concentrations. Currently, the research on H_2_S was actively turned to its signaling role from toxic gas molecules in plants [1]. Signaling molecules, like H_2_S, can be rapidly synthesized and triggered by signaling pathways when plants are subjected to environment stimuli, and even stress, but they maintain homeostasis in plant cellular and subcellular compartments under normal physiological conditions [2]. H_2_S homeostasis in plant cells is strictly controlled by anabolic and catabolic enzymes located in the various subcellular compartments. These metabolic enzymes include L-/D-cysteine desulfhydrase (L/DCD), O-acetylserine (thiol)lyase A1 (OAS-TL A1), and L-cysteine desulfhydrase 1 (DES1, OAS-TL homolog) in the cytosol; sulfite reductase (SiR) and OAS-TL B in chloroplast; and β-cyanoalanine synthase (CAS), OAS-TL C, and nitrogenase Fe-S cluster (NifS) in mitochondria [3]. These enzymes can be activated or inhibited by a specific metabolism, development, and/or environment signals/stress to trigger H_2_S signaling or maintain H_2_S homeostasis in plants [4]. H_2_S, as a novel gasotransmitter, could increase multiple-stress tolerance in plants [5]. In strawberry plants, H_2_S-induced system tolerance to salt and osmotic stress occurs by minimizing the oxidative (i.e., H_2_O_2_) and nitrosative (i.e., NO) stress via the transcriptional regulation of the ROS-scavenging system [6]. Similarly, H_2_S improved the thermotolerance in strawberry plants through regulating the transcription of heat shock proteins (HSPs) and aquaporin (AQP) [7].

Abscisic acid (ABA), a sesquiterpene, is a phytohormone that regulates seed germination, plant growth, development, and response to environmental stress. ABA can be rapidly synthesized under stress conditions, such as extreme temperature (high and low temperature), salt, and drought stress, and thus increases the stress tolerance in plants [8,9]. Therefore, ABA is also known as a stress hormone. In general, ABA can be biosynthesized by the carotenoid pathway (salvage pathway) and the de novo pathway [10,11,12]. In the carotenoid pathway, zeaxanthin epoxidase (ZEP) catalyzes zeaxanthin to all-trans-violaxanthin, which is converted into all-trans-neoxanthin or 9-cis-violaxanthin in plastids. Further, both all-trans-neoxanthin and 9-cis-violaxanthin are converted into xanthoxin by 9-cis epoxycarotenoid dioxygenase (NCED). And then, xanthoxin is exported into cytosol, which is in turn converted into abscisic aldehyde by short-chain dehydrogenase/reductase (SDR). Finally, abscisic aldehyde is oxidized to ABA by abscisic aldehyde oxidase (AAO), which needs a molybdenum cofactor [9,10]. Also, in the de novo pathway, namely the terpenoid synthetic pathway, isopentenyl diphosphate (IPP) and its isomer dimethylallyl diphosphate (DMAPP) are used as precursors to synthesize carotenoids by the mevalonic acid (MVA) pathway via intermediates farnesyl diphosphate (FPP) and methyl erythritol phosphate (MEP) in chloroplasts. Finally, carotenoids can be converted into ABA by the carotenoid pathway. In addition, ABA can be inactivated by the catalysis of ABA 8′-hydroxylases (CYP707A) or glucosyltransferases to form 8′-hydroxyl abscisic acid and ABA-glucosyl ester [11,12]. ABA, as a stress hormone, could elevate the tolerance of plants to multiple abiotic stress [13]. In rice plants, ABA prevented high temperature stress injury by modulating sugar metabolism [14]. Similarly, ABA pretreatment improved the tolerance to drought, salt, cold, and heat stress in rice by enhancing tricarboxylic acid cycle activity and collaborating with stress response proteins [13]. Also, the thermotolerance in rice plants could be increased by ABA via modulating stomatal conductance and tissue temperature [15].

Many studies show that H_2_S and ABA can exert their physiological functions in plants in a synergistic or antagonistic manner. These physiological functions are involved in stomatal movement [16,17], plant growth and development [18], and stress tolerance [18,19]. Though our previous study also showed that ABA can increase the resistance of tobacco cells to heat stress (HS) by triggering H_2_S signaling via activating LCD [20], its physiological and molecular mechanism remains elusive. HS, as a major abiotic stress factor, severely impacts crop plants, including maize yield, with the occurrence of greenhouse effects all over the word [21,22,23,24]. Maize, as the third food crop worldwide, has been planted in Yunnan, China, due to its multiple-stress tolerance and high yield. As mentioned above, H_2_S and ABA alone could improve the multiple-stress tolerance including heat stress tolerance in plants, which has very important significance for agriculture, but the effect of their combined treatment on thermotolerance in maize plants still remains elusive. Hence, understanding the mechanism underlying H_2_S-ABA interaction-induced thermotolerance in maize is important to breed a climate-resilient and thermotolerant variety for maize production. This paper, using maize as the materials, aims to uncover the mechanism underlying H_2_S-ABA interaction-induced thermotolerance in maize seedlings.

## 2. Results

### 2.1. H_2_S and ABA Alone or in Combination Induces Thermotolerance

After HS and, subsequent, recovery at 26 °C for one week, the survival rate of the seedlings irrigated with NaHS and ABA alone, or in combination, was counted. The results show that, compared with the control, the survival rate (SR) of the maize seedlings irrigated with NaHS and ABA alone, or in combination, after HS was significantly improved, the SR was increased to 78%, 82%, and 85%, respectively, from 52% in the control, especially irrigation with NaHS and ABA in combination was more significant among the treatments (Figure 1A). Similarly, before and after HS, the electrolyte leakage (EL, denoting biomembrane integrity) and malondialdehyde (MDA, membrane lipid peroxidation) content in the seedlings irrigated with NaHS and ABA alone, or in combination, were determined. The data indicate that NaHS and ABA alone, or in combination, had no significant effect on the electrolyte leakage (Figure 1B) and MDA content (Figure 1C) before HS. After HS, NaHS and ABA alone, or in combination, obviously remitted the increase in the electrolyte leakage (Figure 1B) and MDA content (Figure 1C) compared with the control, especially NaHS combined with ABA in combination showed a more significant remission, similar to the effect on the survival rate (Figure 1A). These data suggest that H_2_S and ABA alone, or in combination, could induce thermotolerance in maize seedlings.

### 2.2. ABA Increases Endogenous H_2_S Level

To further illustrate the H_2_S-ABA crosstalk, before and after HS, the contents of endogenous H_2_S and ABA, as well as the activities by their metabolic enzymes in seedlings irrigated with NaHS and ABA alone or in combination, NaHS combined with ST or FLU, as well as ABA combined with PAG, HA, or HT, were determined. The data indicate that, before HS, the endogenous H_2_S content was significantly increased by NaHS and ABA alone, or in combination, and the combination resulted in a more obvious increase (Figure 2). Also, NaHS-induced H_2_S was eliminated by the ABA inhibitors ST and FLU, while the ABA-induced H_2_S was weakened by the H_2_S inhibitor PAG, but eliminated by the H_2_S scavengers HA and HT. Similarly, the activity of LCD, DCD, and OAS-TL in the maize seedlings was significantly enhanced by H_2_S and ABA alone, or in combination (Figure 3). However, the gene expression of *ZmLCD1* and *ZmOAS-TL* was not markedly up-regulated by NaHS and ABA alone, or in combination, except NaHS up-regulated *ZmOAS-TL* expression (Figure 4).

After HS, the endogenous H_2_S level in maize seedlings irrigated with NaHS and ABA alone or in combination was significantly increased, especially both in combination were shown to be more effective (Figure 2). Similarly, NaHS-induced H_2_S was removed by the ABA inhibitors ST and FLU; whereas, ABA-induced H_2_S was impaired by the H_2_S inhibitor PAG and scavengers HA and HT, respectively (Figure 2). Analogously, the LCD, DCD, and OAS-TL activity was markedly increased by H_2_S and ABA alone or in combination, except ABA alone for OAS-TL (Figure 3). For gene expression, *ZmLCD1* expression was significantly up-regulated by ABA alone or combined with NaHS, but down-regulated by NaHS alone. Also, *ZmOAS-TL* expression was markedly up-regulated by ABA alone or combined with NaHS, but NaHS had no significant effect on its expression (Figure 4).

### 2.3. H_2_S Increases Endogenous ABA Level

To further study the effect of NaHS on the endogenous ABA level, the content of the ABA and its metabolic enzyme activity in maize seedlings irrigated with NaHS and ABA alone or in combination, NaHS combined with ST and FLU, and ABA combined with PAG, HA, and HT, were measured. Before the HS, irrigation with NaHS, ABA, ST + NaHS, FLU + NaHS had no significant effect on the endogenous ABA level, which was obviously increased by the ABA + NaHS, whereas it was significantly weakened by the combination of ABA with PAG, HA, or HT (Figure 5). Also, the activities of ZEP, NCED, and AAO in seedling mesocotyls were markedly improved by NaHS and ABA alone, or in combination (Figure 6). Similarly, the gene expression of *ZmZEP*, *ZmNCED*, and *ZmAAO* in the mesocotyls of maize seedlings was observably up-regulated by the irrigation with NaHS and ABA alone, or in combination (Figure 7).

After HS, the endogenous ABA content in the mesocotyls of the maize seedlings was signally increased by NaHS and ABA alone or in combination, but impaired by the combination of ABA with PAG, HA, or HT, while the combination of NaHS with ST or FLU had no significant effect on the endogenous ABA level (Figure 5). In addition, the activities of the ABA metabolic enzymes (ZEP, NCED, and AAO) were memorably increased by the irrigation with NaHS and ABA alone or in combination, especially both in combination was more significant (Figure 6). Analogously, the expression of *ZmZEP*, *ZmNCED*, and *ZmAAO* in the mesocotyls of the maize seedlings was dramatically up-regulated by the irrigation with NaHS and ABA alone, or in combination (Figure 7).

### 2.4. H_2_S-ABA Crosstalk Activates ROS-Scavenging System

To further understand the mechanism underlying H_2_S-ABA crosstalk-induced thermotolerance in maize seedlings, the activity of the ROS-scavenging system was analyzed. Before the HS, the results show that the activities of GR, MDHAR, CAT, and DHAR in maize seedlings were dramatically increased by the irrigation with NaHS and ABA alone or in combination (Figure 8, Figure 9 and Figure 10), while APX, POD, and SOD activity was only activated by NaHS combined with ABA, but NaHS and ABA alone had no significant effect on the activity of the three enzymes (Figure 8 and Figure 9). Correspondingly, the contents of the carotenoids and flavone in the maize seedlings were observably augmented by NaHS and ABA alone or in combination (Figure 10 and Figure 11), whereas the AsA and total phenol contents were only augmented by the combination of NaHS with ABA, but NaHS and ABA had no significant difference on the contents of the AsA and total phenol in the maize seedlings (Figure 10 and Figure 11). Similarly, the gene expression of *ZmGR1* and *ZmAPX1* in the mesocotyls of the maize seedlings was significantly up-regulated by NaHS and ABA alone or in combination (Figure 12 and Figure 13), while *ZmCAT*_1_ expression was obviously up-regulated by the irrigation with NaHS alone or combined with ABA, and up-regulation of *ZmSOD*_4_ expression by NaHS and ABA alone was also observed (Figure 12 and Figure 13), but the significant difference from NaHS and ABA alone or in combination on *ZmDHAR* and *ZmMDHAR* expression was not observed, except for the significant effect of both in combination on *ZmMDHAR* expression (Figure 12 and Figure 13). 

After HS, the activities of APX, CAT, POD, and MDHAR in the mesocotyls of the maize seedlings were significantly increased by the irrigation with NaHS and ABA alone or in combination (Figure 7, Figure 8 and Figure 9), while a significant difference from NaHS and ABA alone or in combination on the activity of GR, DHAR, and SOD in maize seedlings was not noted (Figure 7, Figure 8 and Figure 9). Also, the contents of the carotenoids, flavone, and total phenols in the maize seedlings were obviously increased by the irrigation with NaHS and ABA alone or in combination, both in combination increased the AsA content, but NaHS and ABA alone had no significant effect on the AsA (Figure 9 and Figure 10). Correspondingly, the gene expression of *ZmCAT1*, *ZmSOD4*, *ZmGR1*, and *ZmMDHAR* in the maize seedlings was markedly up-regulated by NaHS and ABA alone or in combination (Figure 11 and Figure 12), and *ZmAPX1* and *ZmDHAR* expression was separately up-regulated by NaHS combined with ABA and ABA alone, or in combination with NaHS (Figure 11 and Figure 12). 

### 2.5. H_2_S-ABA Crosstalk Modulates the ROS Level

As mentioned above, H_2_S-ABA crosstalk could activate the activity of the ROS-scavenging system in maize seedlings (Figure 8, Figure 9, Figure 10, Figure 11, Figure 12 and Figure 13). To further explore the effect of H_2_S-ABA crosstalk on the ROS level in maize seedlings, the H_2_O_2_ content and O_2_^.−^ generation rate was detected. Before the HS, the data show that the generation rate for O_2_^.−^ in the mesocotyls of the maize seedlings was significantly decreased by the irrigation with NaHS and ABA alone or in combination, but the content of the H_2_O_2_ was increased through the irrigations (Figure 14). After the HS, the generation rate for O_2_^.−^ in the mesocotyls of the maize seedlings was significantly increased, but the HS-induced increase in the generation rate for O_2_^.−^ was weakened by NaHS and ABA alone or in combination (Figure 14). Also, compared with the control, the H_2_O_2_ content in the mesocotyls of the seedlings was maintained at a low level by NaHS and ABA alone, or in combination (Figure 14). 

### 2.6. Correlation among Indices

After the correlation analysis, the correlation between H_2_S and its metabolic enzymes (LCD, DCD, and OAS-TL) and ABA and its metabolic enzymes (ZEP, NCED, and AAO) is shown in Table 1. The data showed that H_2_S was positively correlated with ABA, ZEP, NCED, and AAO, as well as with ABA, and AAO, and NCED reached significant levels.
ijms-24-12264-t001_Table 1Table 1Correlation analysis between H_2_S and its metabolic enzymes and ABA and its metabolic enzymes. In the table, r represents the correlation coefficient (positive and negative numbers denote positive and negative correlation, respectively), the asterisk (*) and double asterisks (**) indicate significant (*p <* 0.05) and very significant difference (*p <* 0.01), respectively.rH_2_SLCDDCDOAS-TLABA0.672 *0.873 **0.825 **0.821 **ZEP0.5890.724 **0.703 **0.862 **NCED0.727 **0.604 *0.610 *0.661 *AAO0.671 *0.787 **0.614 *0.861 **Correspondingly, ABA was positively associated with LCD, DCD, and OAS-TL, and the correlation reached very significant difference (*p <* 0.01). Also, Table 2 indicates that the survival rate (SR) was positively correlated with tissue viability (TV), and the correlation showed very significant difference (*p <* 0.01), while SR was negatively correlated with electrolyte leakage (EL) and malondialdehyde (MDA), and their correlations showed a very significant difference.
ijms-24-12264-t002_Table 2Table 2Correlation analysis between the survival rate and heat tolerance index. In the table, r represents the correlation coefficient (positive and negative numbers denote positive and negative correlation, respectively), the double asterisks (**) indicate significant (*p <* 0.05) and very significant difference (*p <* 0.01), respectively.rSRELTVMDASR1


EL−0.561 **1

TV0.331 **−0.1381
MDA−0.622 **0.175−0.515 **1In addition, Table 3 implies that a very significant positive correlation (*p <* 0.01) was observed between SR and POD, CAT, GR, APX, and DHAR, while a significant positive correlation (*p <* 0.05) existed between SR and MDHAR.
ijms-24-12264-t003_Table 3Table 3Correlation analysis between the survival rate and antioxidant enzymes. In the table, r represents the correlation coefficient (positive and negative numbers denote positive and negative correlation, respectively), the asterisk (*) and double asterisks (**) indicate significant (*p <* 0.05) and very significant difference (*p <* 0.01), respectively.rSRPODCATSODGRAPXDHARMDHARSR1






POD0.459 **1





CAT0.636 **0.405 **1




SOD0.0850.1140.288 **1



GR0.456 **0.339 **0.374 **0.1341


APX0.424 **0.325 **0.467 **0.307 **0.345 **1

DHAR0.471 **0.387 **0.324 **0.1670.385 **0.267 *1
MDHAR0.245 *0.318 **0.168−0.0490. 348 **0.0160.1031Similarly, in Table 4, a very significant positive correlation exists between SR and AsA, FLA, TP, CAR, and H_2_O_2_, whereas a very significant negative correlation is noted between SR and O_2_^.−^_._
ijms-24-12264-t004_Table 4Table 4Correlation analysis between the survival rate and antioxidants. In the table, r represents the correlation coefficient (positive and negative numbers denote positive and negative correlation, respectively), the asterisk (*) and double asterisks (**) indicate significant (*p <* 0.05) and very significant difference (*p <* 0.01), respectively.rSRAsADHAFLATPCARH_2_O_2_O_2_·^−^SR1






AsA0.303 **1





DHA−0.75−0.731




FLA0.499 **0.241 *−0.1381



TP0.515 **0.306 **0.0170.351 **1


CAR0.513 **0.362 **−0.248 *0.361 **0.399 **1

H_2_O_2_0.582 **0.397 **0.0110.407 **−0.576 **0.467 **1
O_2_·^−^−0.610 **−0.186−0.060−0.417 **−0.370 **−0.446 **−0.383 **1

## 3. Discussion

H_2_S, as a novel signaling molecule, and ABA, as a stress hormone, can exert their physiological functions in an independent or dependent (antagonistic or synergistic) manner in plants [16,19,25]. However, in the development of thermotolerance in plants, whether and how H_2_S interacts with ABA remains elusive. In this paper, using maize seedlings as the materials, we found that H_2_S interacted with ABA by mutually modulating their endogenous levels via metabolic enzyme activity and corresponding gene expression (Figure 2, Figure 3, Figure 4, Figure 5 and Figure 6), and the ROS-scavenging system played an essential role in H_2_S-ABA crosstalk-induced thermotolerance in maize seedlings (Figure 7, Figure 8, Figure 9, Figure 10, Figure 11, Figure 12 and Figure 13).

Generally, the crosstalk among signaling molecules maybe achieve by chemical reaction among the signaling molecules, by modulating the metabolic enzymes, by competing with common target proteins, and/or by regulating some nodes in the signaling pathways, and so on [26,27]. Therefore, some signaling pathways, through signaling crosstalk, can be strengthened, or weakened, and can even trigger a novel signal pathway [26,27]. In this paper, irrigation with NaHS up-regulated the gene expression of the ABA metabolic enzymes (*ZmZEP*, *ZmNCED1*, and *ZmAAO*) in the maize seedlings, which in turn increased the level of endogenous ABA under both non-HS and HS conditions (Figure 6 and Figure 7). Also, NaHS-induced ABA was weakened by the ABA inhibitors ST and FLU, respectively (Figure 6 and Figure 7), indicating that H_2_S could trigger ABA signaling by activating its metabolic pathways. Similarly, root irrigation with ABA up-regulated the gene expression of the H_2_S metabolic enzymes (*ZmLCD1* and *ZmOAS-TL*) in the maize seedlings, followed by increasing the level of endogenous H_2_S under both non-HS and HS conditions (Figure 2 and Figure 3). In addition, ABA-induced H_2_S was impaired by the H_2_S inhibitor PAG and scavengers HA and HT, respectively (Figure 2 and Figure 3), indicating that ABA could trigger H_2_S signaling by modulating its metabolic enzymes. Also, Pearson correlation analysis showed that H_2_S and its metabolic enzymes LCD, DCD, and OAS-TL were positive correlation with ABA and its metabolic enzymes ZEP, NCED, and AAO, and their correlation reached a significant (for ZEP and AAO) and very significant (for NCED) difference (Table 1). Similarly, in wheat and rice plants, H_2_S regulated the formation of thermotolerance by interacting with ABA, melatonin, ETH, and NO [28,29,30]. These results suggest that the crosstalk between H_2_S and ABA occurred in the development of plant thermotolerance by mutually modulating the metabolic enzymes.

In addition, numerous studies show that H_2_S and ABA alone or in combination could increase the thermotolerance in plants, but their detailed mechanisms were not completely clear [20,31,32]. In this paper, the thermotolerance in maize seedlings was significantly improved by increasing the survival rate and decreasing the electrolyte leakage and MDA accumulation after the seedlings were irrigated with NaHS and ABA alone or in combination, particularly in combination (Figure 1). Also, Pearson correlation analysis indicated that the survival rate was positively correlated with tissue viability, negatively correlated with electrolyte leakage and MDA accumulation, and their correlation reached a significant difference (Table 2). These results further support the fact that H_2_S-ABA crosstalk induces thermotolerance in maize seedlings. 

Oxidative stress is the major heat injury in plants due to the excessive accumulation of ROS under HS conditions [23,33,34,35]. Correspondingly, the acquirement of thermotolerance in plants and the ROS-scavenging system goes hand in hand [36,37,38,39,40,41]. In this paper, HS promoted the generation rate of O_2_^.−^ in the mesocotyls of the maize seedlings (Figure 14), indicating HS-triggered oxidative stress. Interestingly, HS-promoted O_2_^.−^ production was significantly impaired by NaHS and ABA alone or in combination, especially in combination (Figure 14). To further understand the mechanisms underlying H_2_S-ABA crosstalk-induced thermotolerance and the mitigative effect of NaHS and ABA alone or in combination on oxidative stress in maize seedlings, the ROS-scavenging system activity was analyzed. Before the HS, the enhanced ROS-scavenging system activity by NaHS and ABA alone or in combination laid the foundation for the acquirement of subsequent thermotolerance in the maize seedlings (Figure 8, Figure 9, Figure 10, Figure 11, Figure 12 and Figure 13). After irrigation with NaHS and ABA alone or in combination, the activity of the ROS-scavenging enzymes, such as CAT, APX, POD, and MDHAR, as well as the content of the non-enzymatic antioxidants, such as AsA, carotenoids, flavone, and total phenols, in the maize seedlings were significantly enhanced, particularly when irrigating with NaHS in combination with ABA (Figure 8, Figure 9, Figure 10, Figure 11, Figure 12 and Figure 13). Correspondingly, the gene expression of the ROS-scavenging enzymes in the mesocotyls of the maize seedlings, such as *ZmCAT1*, *ZmAPX*, *ZmSOD4*, and *ZmMDHAR*, was markedly up-regulated by the irrigation with NaHS and ABA alone or in combination, especially in combination (Figure 8, Figure 9, Figure 10, Figure 11, Figure 12 and Figure 13). 

In HS situations, the ROS-scavenging system plays a key role in the development of thermotolerance in plants [42,43,44,45,46,47]. As mentioned above, before HS, the enhanced ROS-scavenging system by H_2_S and ABA laid the foundation for the formation of the subsequent thermotolerance (Figure 8, Figure 9, Figure 10, Figure 11, Figure 12 and Figure 13). Analogously, under HS conditions, compared with the control, the ROS-scavenging enzymes (i.e., CAT, APX, POD, and MDHAR) activities and the non-enzymatic antioxidants (i.e., AsA, carotenoids, flavone, and total phenols) contents in the mesocotyls of the maize seedlings were significantly enhanced by the irrigation with NaHS and ABA alone or in combination, especially the combination showed more physiological effects (Figure 8, Figure 9 and Figure 10). Accordingly, the expression of the ROS-scavenging enzyme genes (i.e., *ZmCAT1*, *ZmAPX*, *ZmSOD4*, and *ZmMDHAR*) in the mesocotyls of the maize seedlings was obviously up-regulated by NaHS and ABA alone or in combination, particularly NaHS combined with ABA had the most significantly molecular effect among the irrigations (Figure 12 and Figure 13). In addition, Pearson correlation analysis suggested that the survival rate was positively related to the activity of the ROS-scavenging enzymes, POD, CAT, SOD, GR, APX, DHAR, and MDHAR, and their correlation showed a significant (for MDHAR) and very significant (for POD, CAT, GR, APX, and DHAR) level (Table 3). 

Similarly, the survival rate and the ROS-scavengers AsA, flavone, total phenols, and carotenoids had a positive correlation, while a negative correlation for the survival rate and O_2_^.−^ was observed (Table 4). These indicate that the ROS-scavenging system in maize seedlings could be enhanced by the irrigation with NaHS and ABA alone or in combination under HS conditions, which in turn reduced the accumulation of ROS (Figure 14), thus mitigating the oxidative stress induced by HS and improving the thermotolerance in maize seedlings (Figure 1). As discussed above, H_2_S initiated the formation of thermotolerance by interplaying with ABA, melatonin, ETH, and NO in wheat and rice plants via antioxidants, redox homeostasis, osmolytes, and photosynthetic metabolism [28,29,30], further supporting the essential role of H_2_S-ABA crosstalk in plant thermotolerance development by the ROS-scavenging system. 

## 4. Materials and Methods

### 4.1. Seedling Culture and Treatment

In this study, maize (*Zea mays* L., cv. Chenguang No. 2) seeds were purchased from the Shiling Seed Company, China, as materials. The healthy seeds were selected and sterilized in 5% sodium hypochlorite solution for 10 min, and then washed neatly and soaked at 26 °C for 12 h in distilled water. The soaked seeds were germinated at 26 °C for 2.5 d on six-layer filter papers watered with distilled water in trays with covers (at least 200 seeds per tray). The 2.5 d old seedlings were divided into nine groups and then separately irrigated with 75 mL of the following solutions for 12 h: (1) distilled water (control, CK), (2) 500 μM NaHS, (3) 50 μM ABA, (4) 50 μM ABA + 500 μM NaHS (ABA + NaHS), (5) 500 μM sodium tungstate + 500 μM NaHS (ST + NaHS), (6) 50 μM fluoridone + 500 μM NaHS (FLU + NaHS), (7) 500 μM propargylglycine + 50 μM ABA (PAG + ABA), (8) 500 μM hydroxylamine + 50 μM ABA (HA + ABA), and (9) 500 μM hypotaurine + 50 μM ABA (HT + ABA). Among the chemicals, NaHS, PAG, HA, and HT are H_2_S donors, inhibitors, and scavengers [29,48], while ST and FLU are ABA inhibitors [49]. The suitable concentrations were dated according to the preliminary experiments and previous references [48,50,51]. And then, the irrigated seedlings were subjected to HS at 46 °C for 16 h. Finally, before and after HS, the mesocotyls (which is the most sensitive organ that responds to HS [52]) of the seedlings were selected and used to determine the physiological, biochemical, and molecular parameters according to the following methods, respectively.

### 4.2. Measurement of Thermotolerance Parameters

To study the effect of H_2_S-ABA crosstalk on thermotolerance in maize seedlings, after HS and recovery, the survival rate (SR), tissue viability (TV), biomembrane peroxidation, and electrolyte leakage (EL) of the seedlings irrigated with NaHS and ABA alone or in combination were separately measured, according to the methods reported by Wang et al. [53]. The survival rate and electrolyte leakage were expressed in %, while the biomembrane peroxidation was expressed as nmol g^−1^ fresh weight (FW), respectively. 

### 4.3. Analysis of H_2_S Content and Its Metabolic Enzymes

To investigate the effect of ABA on H_2_S content and its metabolic enzyme activity, the seedlings were irrigated with NaHS alone or combined with the ABA inhibitors ST and FLU, as well as ABA alone or combined with the H_2_S scavenger HT and the inhibitors HA and PAG. After irrigation and HS, the H_2_S content and activity of LCD, DCD, and OAS-TL in seedling mesocotyls were analyzed, as per the procedures by Ye et al. [54]. The H_2_S content was expressed in nmol g^−1^ FW, whereas the LCD, DCD, and OAS-TL activities were indicated as nmol min^−1^ mg^−1^ protein. The content of the soluble proteins in the seedling mesocotyls was analyzed using the Bradford method [55], using bovine serum albumin as the standard sample.

### 4.4. Determination of ABA Content and Its Metabolic Enzymes

Similarly, to explore the effect of H_2_S on the ABA content and its metabolic enzyme activity, the seedlings were irrigated with NaHS alone or combined with the ABA inhibitors ST and FLU, as well as ABA alone or combined with the H_2_S scavenger HT and the inhibitors HA and PAG. After irrigation and HS, the ABA content and activity of ZEP, NCED, and OAA in seedling mesocotyls were determined, as per the procedures in the instruction book for the kits. The ABA content was expressed in μg g^−1^ FW, whereas the ZEP, NCED, and OAA activities were expressed in nmol min^−1^ mg^−1^ protein. 

### 4.5. Estimation of ROS-Scavenging System

To further understand the mechanism underlying H_2_S-ABA crosstalk-induced thermotolerance in maize seedlings, after irrigation and HS, the ROS-scavenging system activity of the seedlings irrigated with NaHS and ABA alone or in combination were estimated according to Elavarthi et al. [56] and Wang et al. [53]. The activity of catalase (CAT), guaiacol peroxidase (POD), superoxide dismutase (SOD), ascorbate peroxidase (APX), glutathione reductase (GR), monodehydroascorbate reductase (MDHAR), and dehydroascorbate reductase (DHAR) was calculated as the millimolar extinction coefficient (mM^−1^ cm^−1^) at 40 (240 nm), 26.6 (470 nm), 2.8 (290 nm), 21.6 (560 nm), 6.2 (340 nm), 14.0 (265 nm), and 14.0 (265 nm), respectively. Their activities were indicated as nmol min^−1^ mg^−1^ protein, except SOD which was indicted as μmol min^−1^ mg^−1^ protein. 

Similarly, the non-enzymatic ROS-scavenging system, namely ascorbic acid (AsA), carotenoids (CAR), flavanone (FLA), and total phenols (TP) in the seedling mesocotyls were extracted and estimated as per the reports by Elavarthi et al. [56] and Wang et al. [53]. Their corresponding contents were expressed in μmol g^−1^ FW, μg g^−1^ FW, and nmol g^−1^ FW.

### 4.6. Determination of ROS

To estimate the effect of H_2_S-ABA crosstalk on the ROS level, after irrigation and HS, the generation rate for the superoxide radical (O_2_^.−^) and hydrogen peroxide (H_2_O_2_) content in seedlings irrigated with NaHS and ABA alone or in combination was determined according to the titanium sulfate method [53] and Na,3′-[1-[(phenylamino)-carbonyl]-3, 4-tetrazolium] (4-methoxy-6-nitro) benzene sulfonic acid hydrate (XTT) method [52]. The O_2_^.−^ production and H_2_O_2_ content were calculated using the millimolar extinction coefficient at 21.6 and 0.28 mM^−1^ cm^−1^, and expressed as nmol min^−1^ g^−1^ FW and nmol g^−1^ FW, respectively.

### 4.7. Quantification of Gene Expression

To further explore the effect of H_2_S-ABA crosstalk on the gene expression of the H_2_S- and ABA-metabolic enzymes and the enzymatic ROS-scavenging system, after irrigation and HS, the genes *ZmLCD1*, *ZmOAS-TL*, *ZmZEP*, *ZmNCED*, *ZmOAA*, *ZmCAT1*, *ZmSOD4*, *ZmGR1*, *ZmAPX1*, *ZmDHAR*, and *ZmMDHAR* in the seedling mesocotyls were quantified by qRT-PCR (using *Zea mays* beta-5 tubulin (*ZmTUB*) as an internal reference), and the relative expression level was calculated using 2^−∆∆CT^ [57]. For the gene primers refer to Appendix A. 

### 4.8. Statistical Analysis

The experiments were a random design and the significance analysis between the data was performed as a one-way analysis of variance (ANOVA) and Duncan multiple-range test at the 0.05 level. The data in the figures represent mean ± standard error (SE, *n* = 5), and the same and different letters indicate the insignificant and significant difference, respectively. Also, to further estimate the correlation among the parameters, the correlation between H_2_S and its metabolic enzymes and ABA and its metabolic enzymes, as well as SR and the thermotolerance index, antioxidant enzymes, non-enzymatic antioxidant, and ROS was analyzed using Pearson correlation in SigmaPlot 25 software. In the tables, r represents the correlation coefficient (positive and negative numbers denote positive and negative correlation, respectively), and the asterisk (*) and double asterisks (**) indicate significant (*p <* 0.05) and very significant difference (*p <* 0.01), respectively.

## 5. Conclusions

In sum, the crosstalk between H_2_S and ABA in maize seedlings before and after HS was found to modulate the activity and gene expression of metabolic enzymes related to H_2_S (LCD, DCD, and OAS-TL, as well as *ZmLCD1* and *ZmOAS-TL*) and ABA (ZEP, NCED, and AAO, as well as *ZmZEP*, *ZmNCED*, and ZmAAO) biosynthesis. The H_2_S-ABA crosstalk induced the thermotolerance in maize seedlings by improving the survival rate (SR) and tissue viability (TV), as well as relieving electrolyte leakage (EL) and MDA accumulation. 

Also, the ROS-scavenging system was enhanced by NaHS and ABA alone or in combination via activation of the enzymatic system (i.e., CAT, APX, GR, POD, SOD, DHAR, and MDHAR) and the non-enzymatic system (i.e., AsA, CAR, FLA, and TP), as well as the up-regulation of the gene expression of the corresponding enzymes (i.e., *ZmCAT1*, *ZmSOD4*, *ZmGR1*, *ZmAPX1*, *ZmDHAR*, and *ZmMDHAR*) in maize seedlings before and after HS. These data indicate the essential role of the ROS-scavenging system in H_2_S-ABA crosstalk-induced thermotolerance in maize seedlings, which lays the foundation for breeding a heat-resilient crop variety, and are very important for maize production.

## Figures and Tables

**Figure 1 ijms-24-12264-f001:**
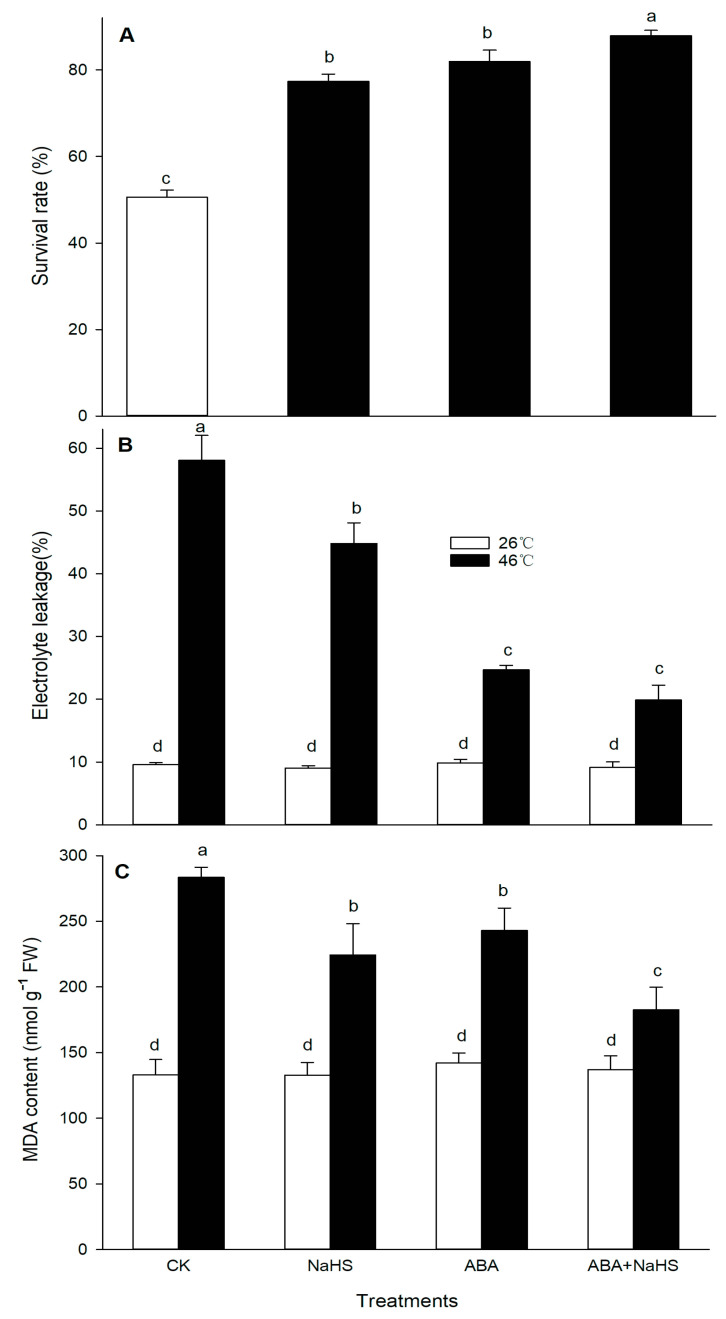
Effect of the irrigation with NaHS and abscisic acid (ABA) alone or in combination on the survival rate (**A**), electrolyte leakage (**B**), and malondialdehyde content (**C**) in maize seedlings under non-heat stress (non-HS) and HS conditions. The significance analysis between the data was performed as a one-way analysis of variance (ANOVA) and Duncan multiple-range test at the 0.05 level. The data in the figures represent the mean ± standard error (SE, *n* = 5), the same and different letters indicate insignificant and significant difference, respectively.

**Figure 2 ijms-24-12264-f002:**
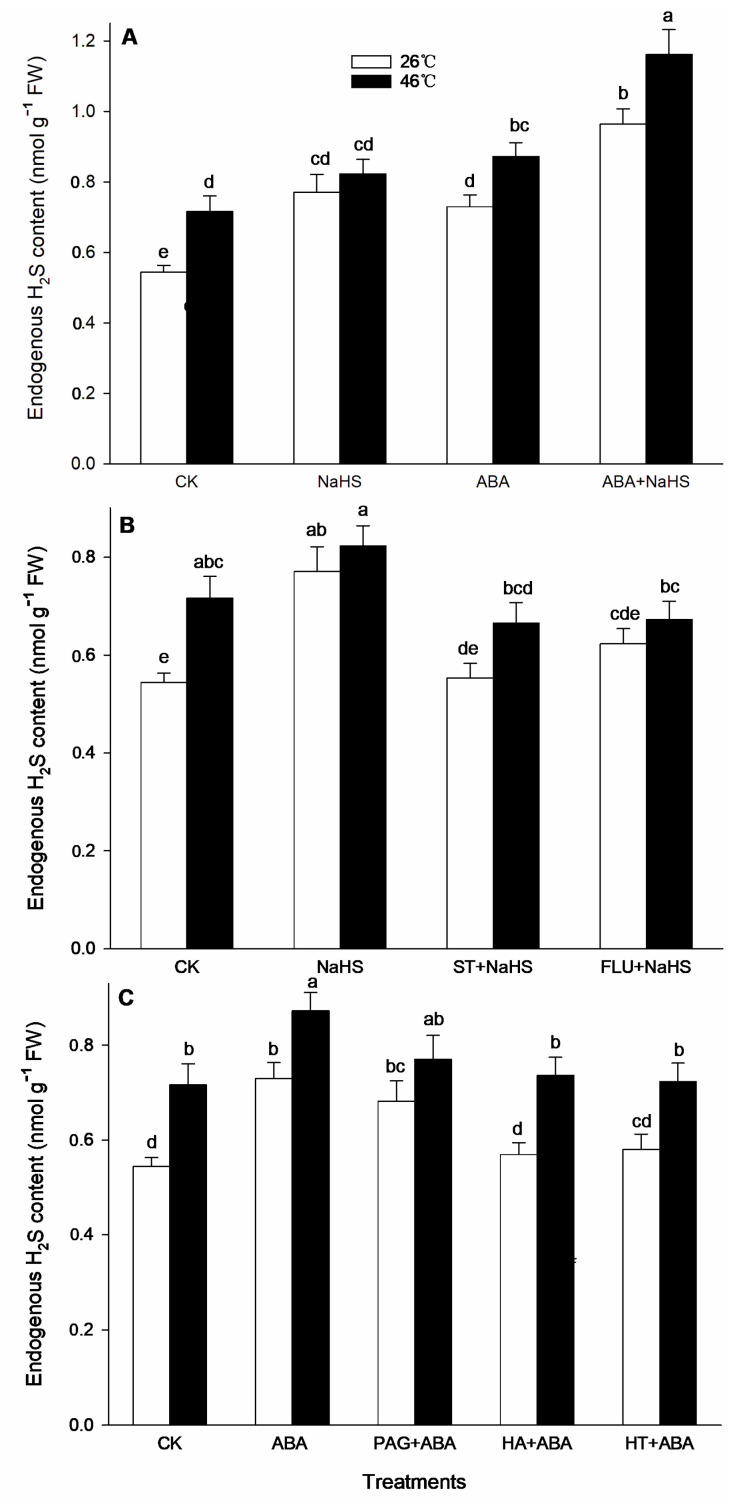
Effect of the irrigation with NaHS and abscisic acid (ABA) alone or in combination (**A**), NaHS alone or in combination with sodium tungstate (ST) or fluoridone (FLU) (**B**), and ABA alone or in combination with hydroxylamine (HA) or hypotaurine (HT) (**C**), on endogenous hydrogen sulfide (H_2_S) content in maize seedlings under non-heat stress (non-HS) and HS conditions. The significance analysis between the data was performed as a one-way analysis of variance (ANOVA) and Duncan multiple-range test at the 0.05 level. The data in the figures represent mean ± standard error (SE, *n* = 5), the same and different letters indicate the insignificant and significant difference, respectively.

**Figure 3 ijms-24-12264-f003:**
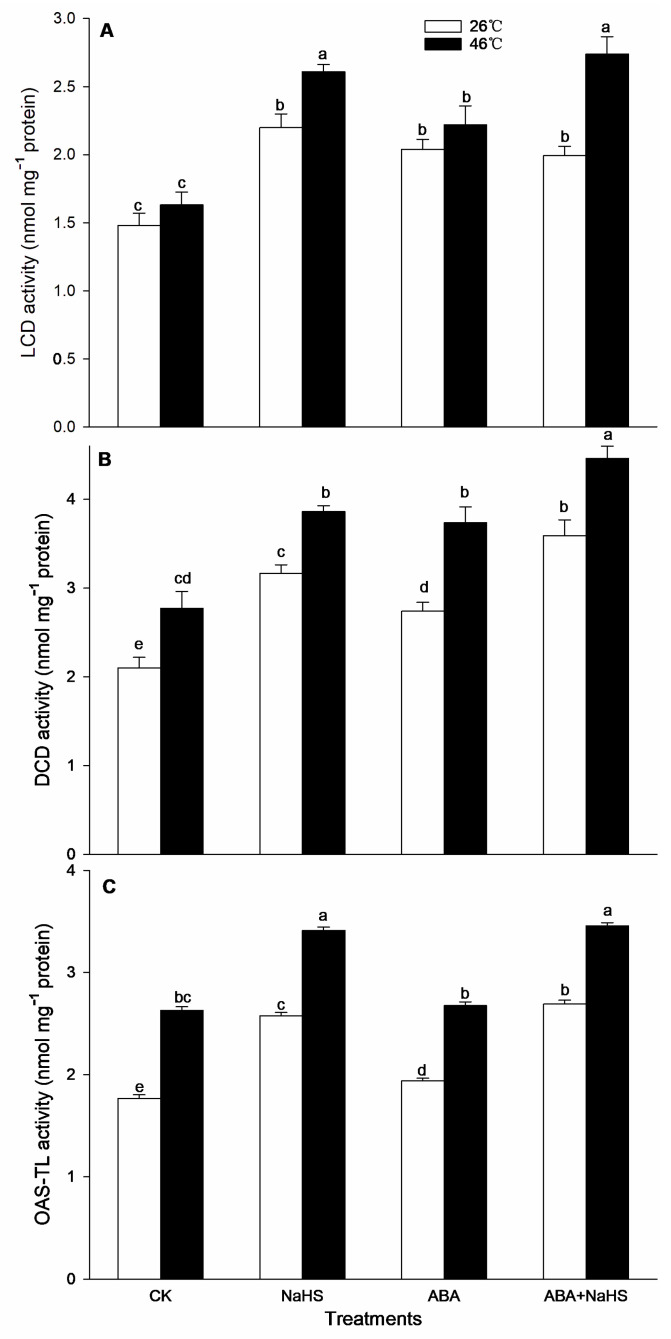
Effect of the irrigation with NaHS and abscisic acid (ABA) alone or in combination on the activity of L-cysteine desulfhydrase (LCD) (**A**), D-cysteine desulfhydrase (DCD) (**B**), and O-acetylserine (thiol)lyase (OAS-TL) (**C**), in maize seedlings under non-heat stress (non-HS) and HS conditions. The significance analysis between the data was performed as a one-way analysis of variance (ANOVA) and Duncan multiple-range test at the 0.05 level. The data in the figures represent mean ± standard error (SE, *n* = 5), the same and different letters indicate the insignificant and significant difference, respectively.

**Figure 4 ijms-24-12264-f004:**
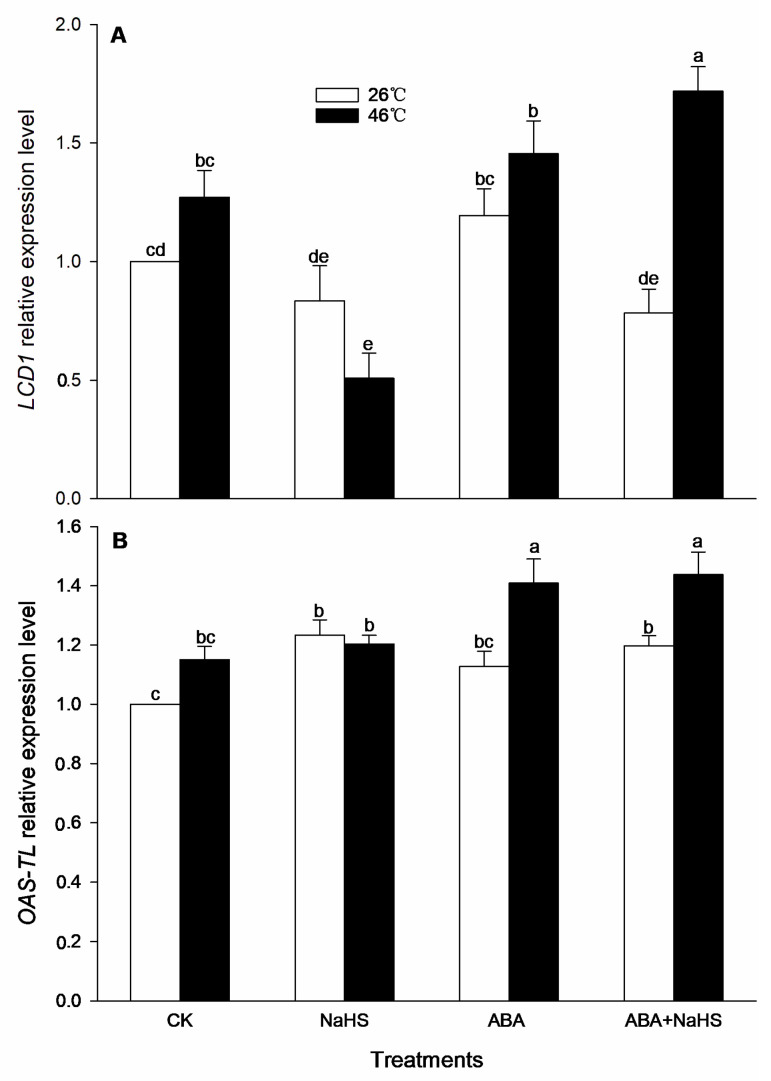
Effect of the irrigation with NaHS and abscisic acid (ABA) alone or in combination on the gene expression of *LCD1* (**A**) and *OAS*-*TL* (**B**) in maize seedlings under non-heat stress (non-HS) and HS conditions. The significance analysis between the data was performed as a one-way analysis of variance (ANOVA) and Duncan multiple-range test at the 0.05 level. The data in the figures represent mean ± standard error (SE, *n* = 5), the same and different letters indicate the insignificant and significant difference, respectively.

**Figure 5 ijms-24-12264-f005:**
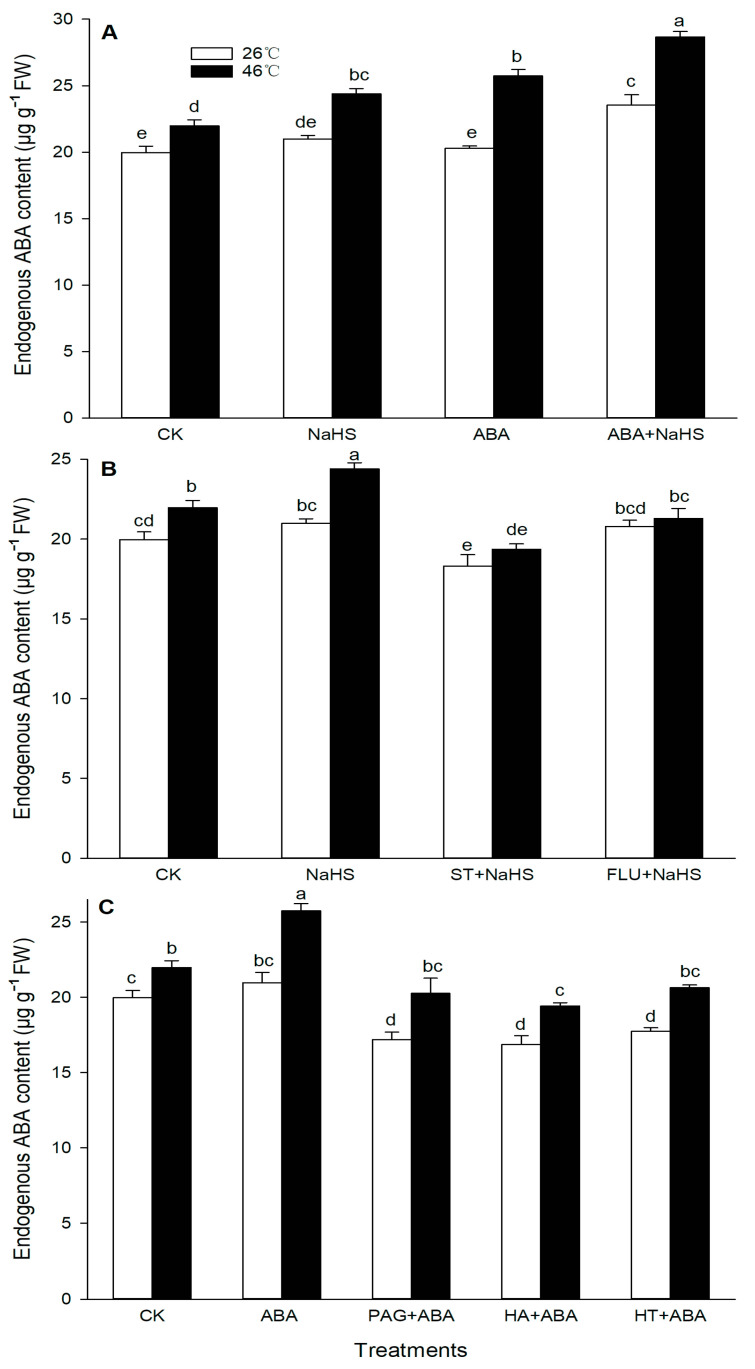
Effect of the irrigation with NaHS and abscisic acid (ABA) alone or in combination (**A**), NaHS alone or in combination with sodium tungstate (ST) or fluoridone (FLU) (**B**), and ABA alone or in combination with hydroxylamine (HA) or hypotaurine (HT) (**C**), on the endogenous ABA content in maize seedlings under non-heat stress (non-HS) and HS conditions. The significance analysis between the data was performed as a one-way analysis of variance (ANOVA) and Duncan multiple-range test at the 0.05 level. The data in the figures represent mean ± standard error (SE, *n* = 5), the same and different letters indicate the insignificant and significant difference, respectively.

**Figure 6 ijms-24-12264-f006:**
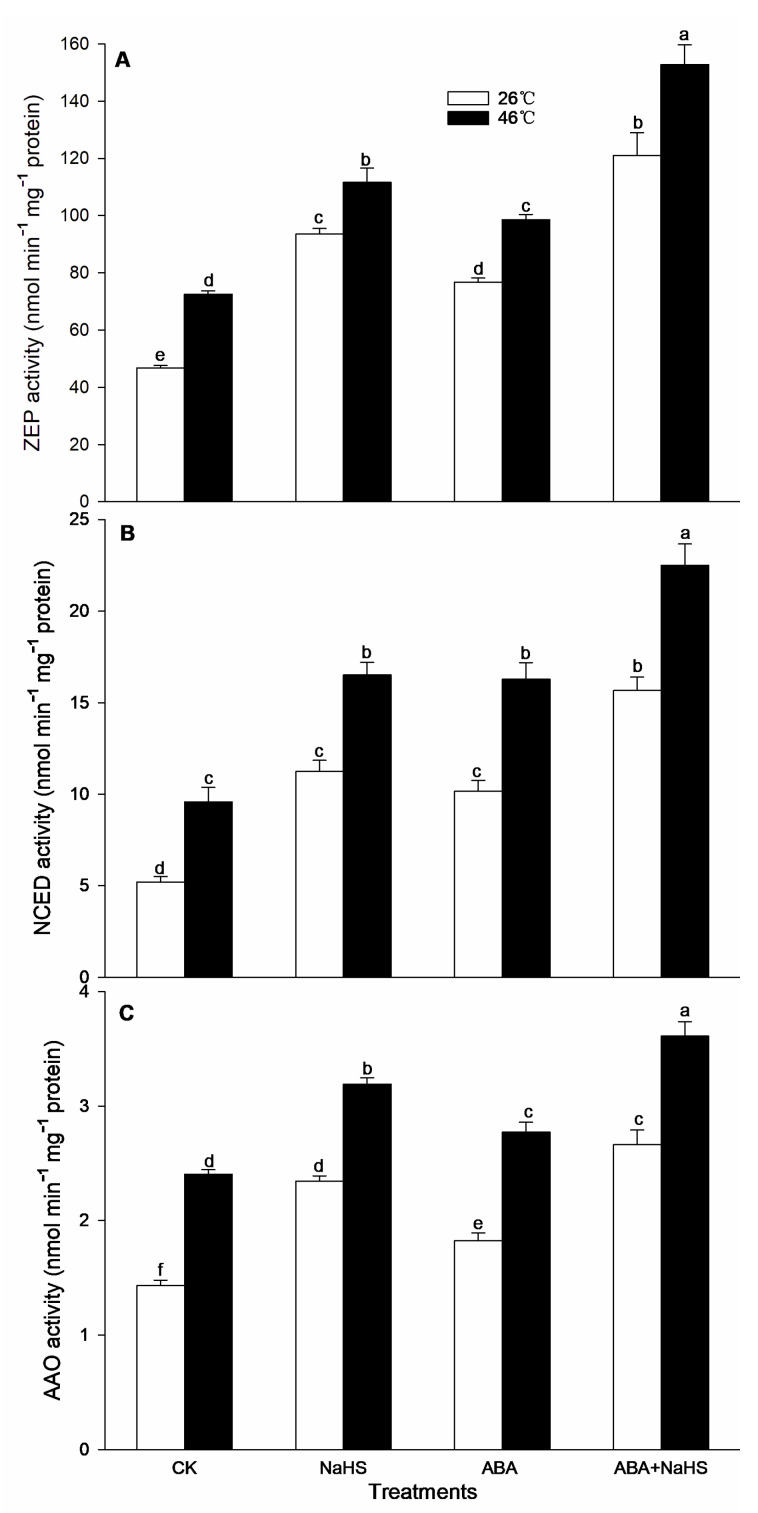
Effect of the irrigation with NaHS and abscisic acid (ABA) alone or in combination on the activity of zeaxanthin epoxidase (ZEP) (**A**), 9-cis epoxycarotenoid dioxygenase (NCED) (**B**), and abscisic aldehyde oxidase (AAO) (**C**) in maize seedlings under non-heat stress (non-HS) and HS conditions. The significance analysis between the data was performed as a one-way analysis of variance (ANOVA) and Duncan multiple-range test at the 0.05 level. The data in the figures represent mean ± standard error (SE, *n* = 5), the same and different letters indicate the insignificant and significant difference, respectively.

**Figure 7 ijms-24-12264-f007:**
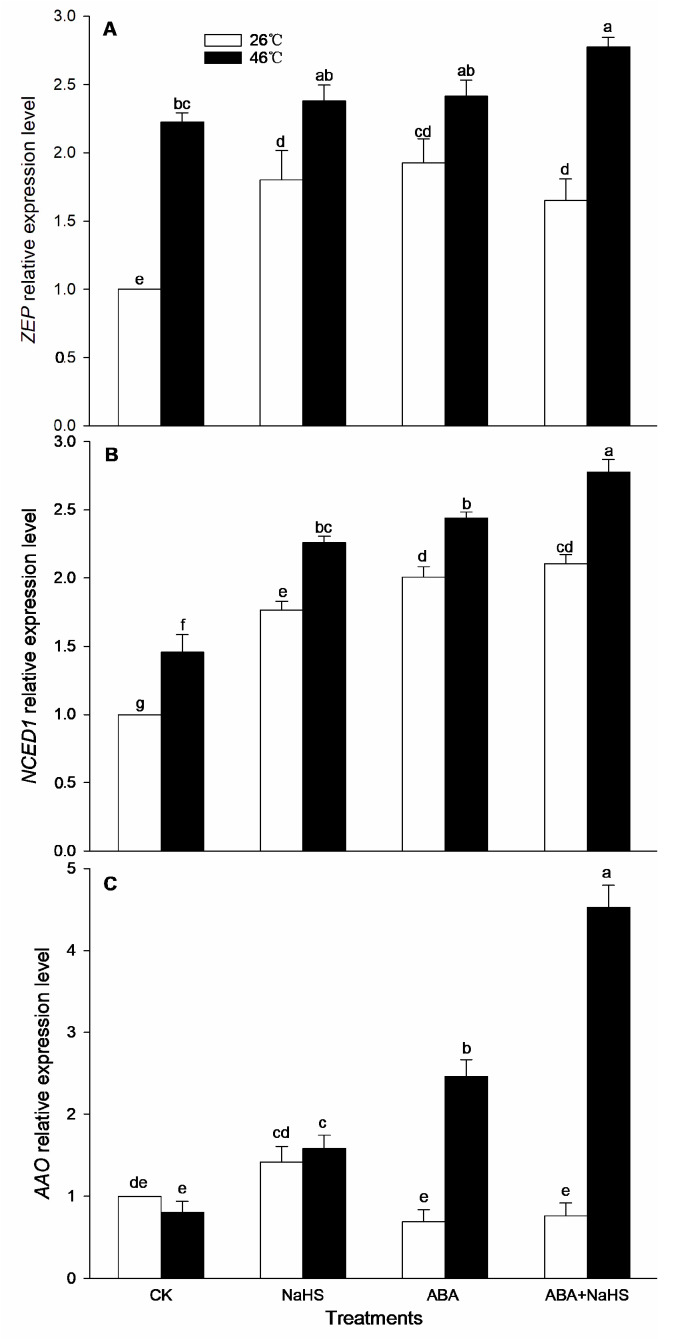
Effect of the irrigation with NaHS and abscisic acid (ABA) alone or in combination on the gene expression of *ZEP* (**A**), *NCED1* (**B**), and *AAO* (**C**) in maize seedlings under non-heat stress (non-HS) and HS conditions. The significance analysis between the data was performed as a one-way analysis of variance (ANOVA) and Duncan multiple-range test at the 0.05 level. The data in the figures represent mean ± standard error (SE, *n* = 5), the same and different letters indicate the insignificant and significant difference, respectively.

**Figure 8 ijms-24-12264-f008:**
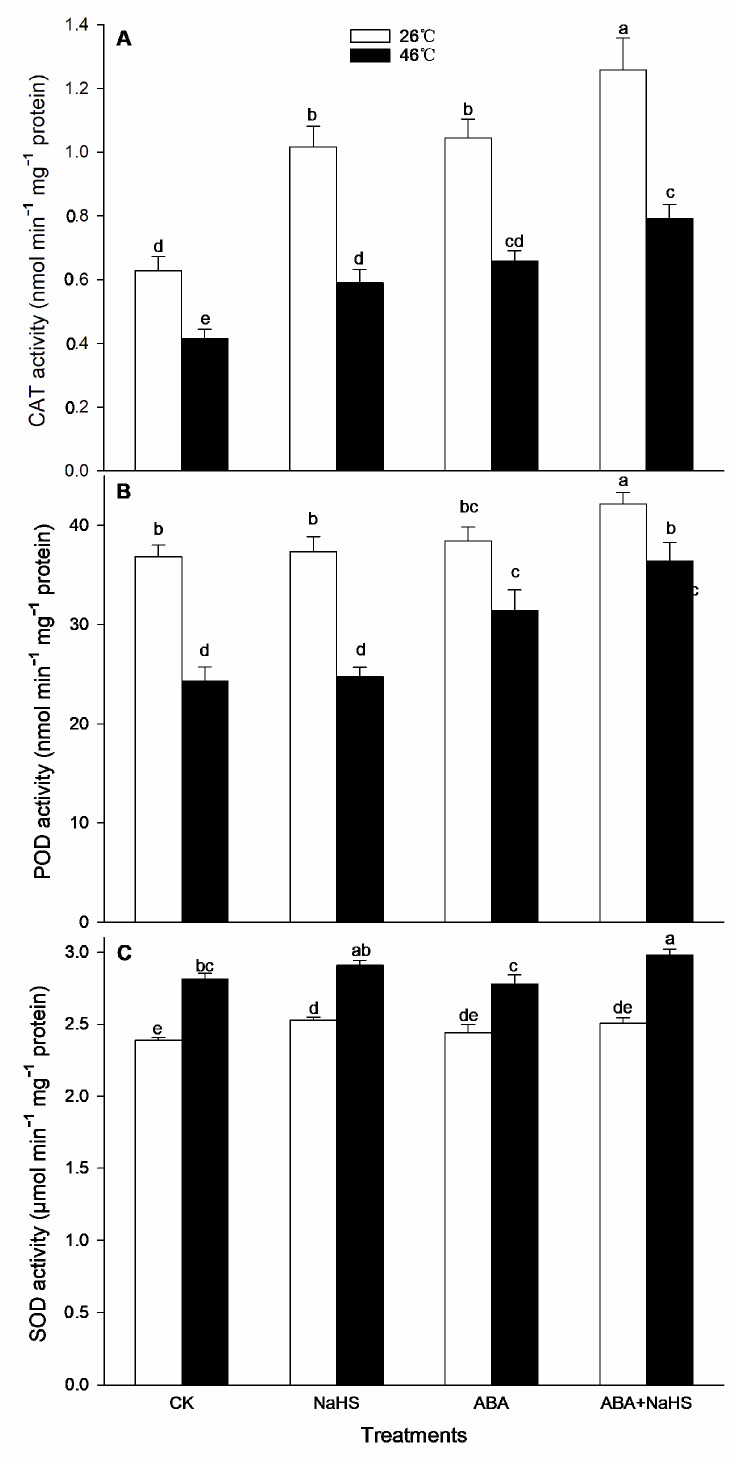
Effect of the irrigation with NaHS and abscisic acid (ABA) alone or in combination on the activity of catalase (CAT) (**A**), peroxidase (POD), (**B**), and superoxide dismutase (SOD) (**C**) in maize seedlings under non-heat stress (non-HS) and HS conditions. The significance analysis between the data was performed as a one-way analysis of variance (ANOVA) and Duncan multiple-range test at the 0.05 level. The data in the figures represent mean ± standard error (SE, *n* = 5), the same and different letters indicate the insignificant and significant difference, respectively.

**Figure 9 ijms-24-12264-f009:**
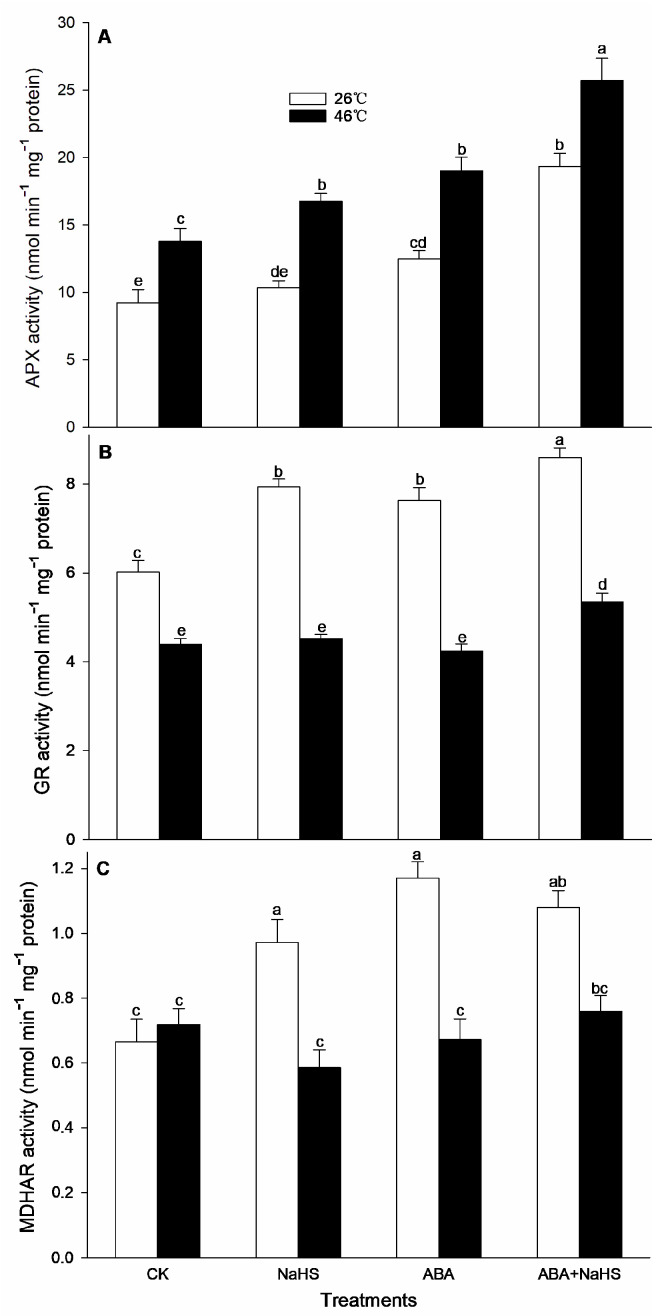
Effect of the irrigation with NaHS and abscisic acid (ABA) alone or in combination on the activity of ascorbate peroxidase (APX) (**A**), glutathione reductase (GR) (**B**), and monodehydroascorbate reductase (MDHAR) (**C**) in maize seedlings under non-heat stress (non-HS) and HS conditions. The significance analysis between the data was performed as a one-way analysis of variance (ANOVA) and Duncan multiple-range test at the 0.05 level. The data in the figures represent mean ± standard error (SE, *n* = 5), the same and different letters indicate the insignificant and significant difference, respectively.

**Figure 10 ijms-24-12264-f010:**
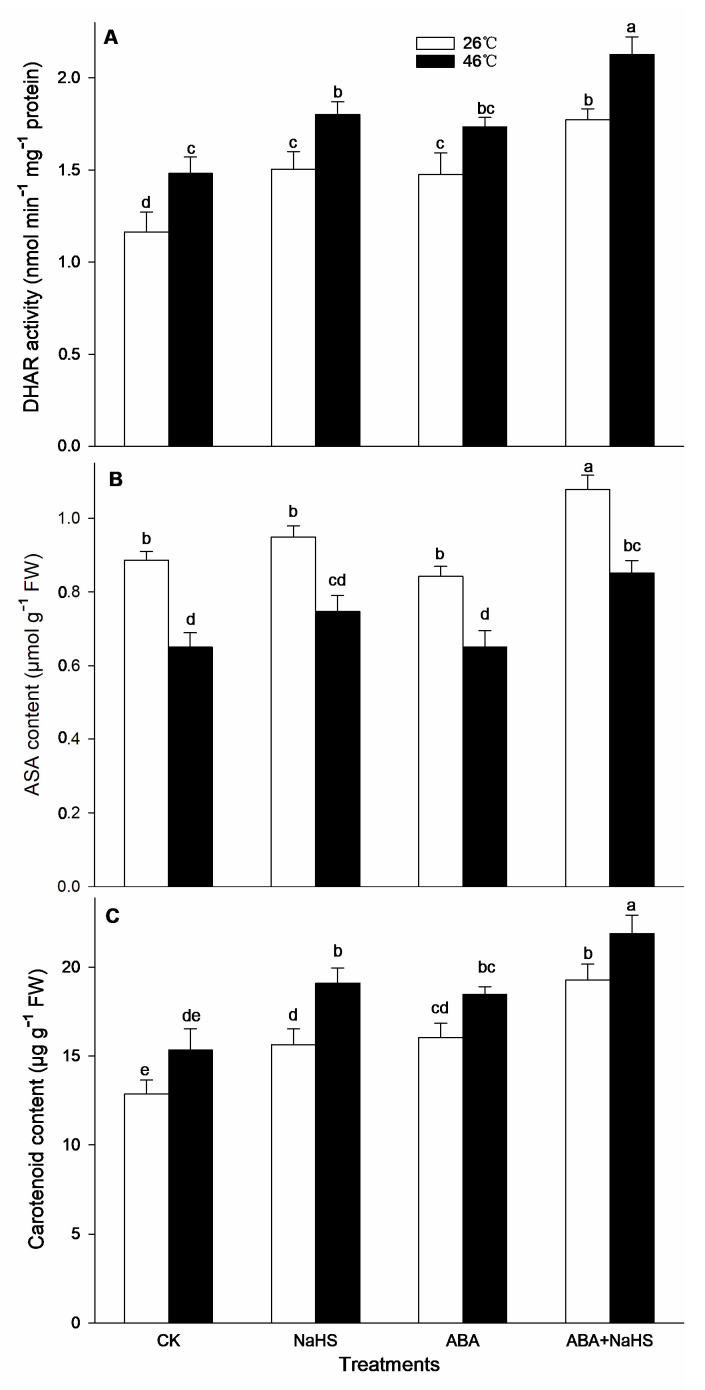
Effect of the irrigation with NaHS and abscisic acid (ABA) alone or in combination on the activity of dehydroascorbate reductase (DHAR) (**A**) and the content of ascorbic acid (ASA) (**B**) and carotenoids (**C**) in maize seedlings under non-heat stress (non-HS) and HS conditions. The significance analysis between the data was performed as a one-way analysis of variance (ANOVA) and Duncan multiple-range test at the 0.05 level. The data in the figures represent mean ± standard error (SE, *n* = 5), the same and different letters indicate the insignificant and significant difference, respectively.

**Figure 11 ijms-24-12264-f011:**
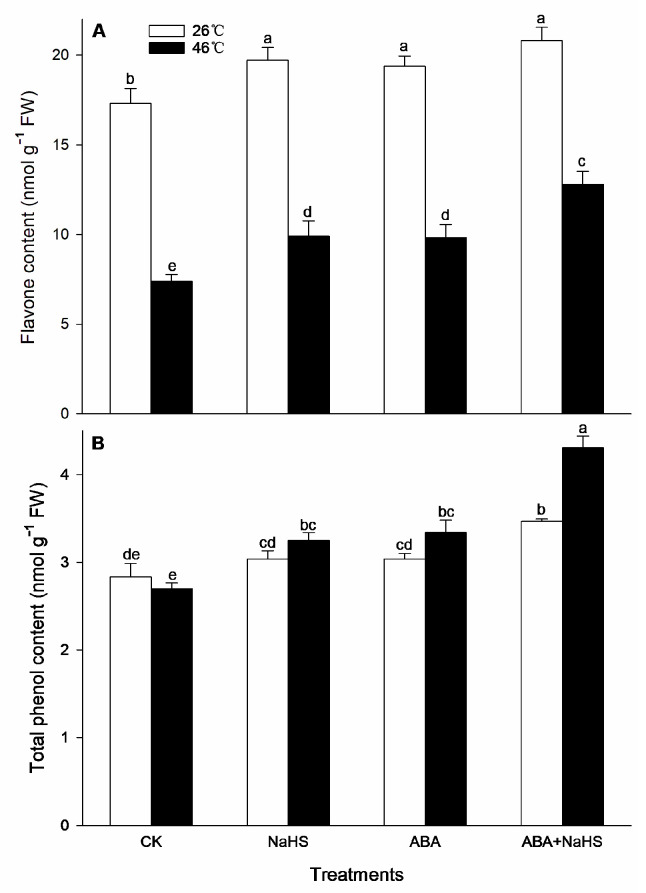
Effect of the irrigation with NaHS and abscisic acid (ABA) alone or in combination on the content of flavone (**A**) and total phenols (**B**) in maize seedlings under non-heat stress (non-HS) and HS conditions. The significance analysis between the data was performed as a one-way analysis of variance (ANOVA) and Duncan multiple-range test at the 0.05 level. The data in the figures represent mean ± standard error (SE, *n* = 5), the same and different letters indicate the insignificant and significant difference, respectively.

**Figure 12 ijms-24-12264-f012:**
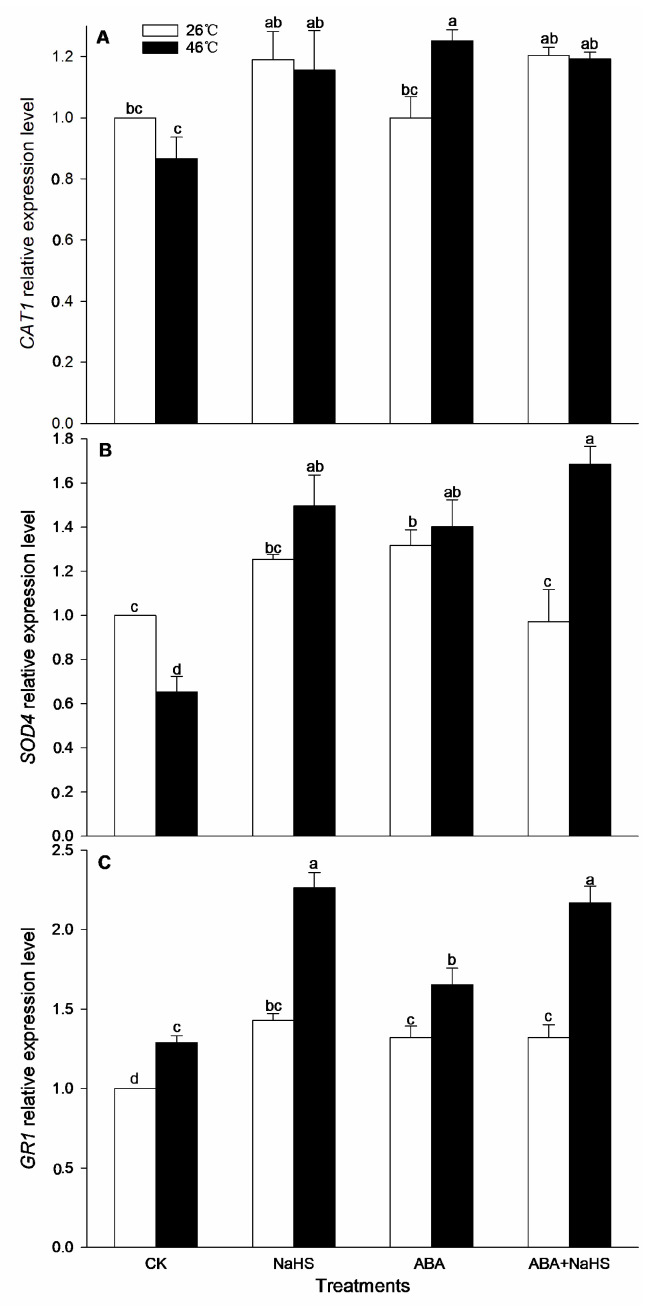
Effect of the irrigation with NaHS and abscisic acid (ABA) alone or in combination on the gene expression of *CAT1* (**A**), *SOD4* (**B**), and *GR1* (**C**) in maize seedlings under non-heat stress (non-HS) and HS conditions. The significance analysis between the data was performed as a one-way analysis of variance (ANOVA) and Duncan multiple-range test at the 0.05 level. The data in the figures represent mean ± standard error (SE, *n* = 5), the same and different letters indicate the insignificant and significant difference, respectively.

**Figure 13 ijms-24-12264-f013:**
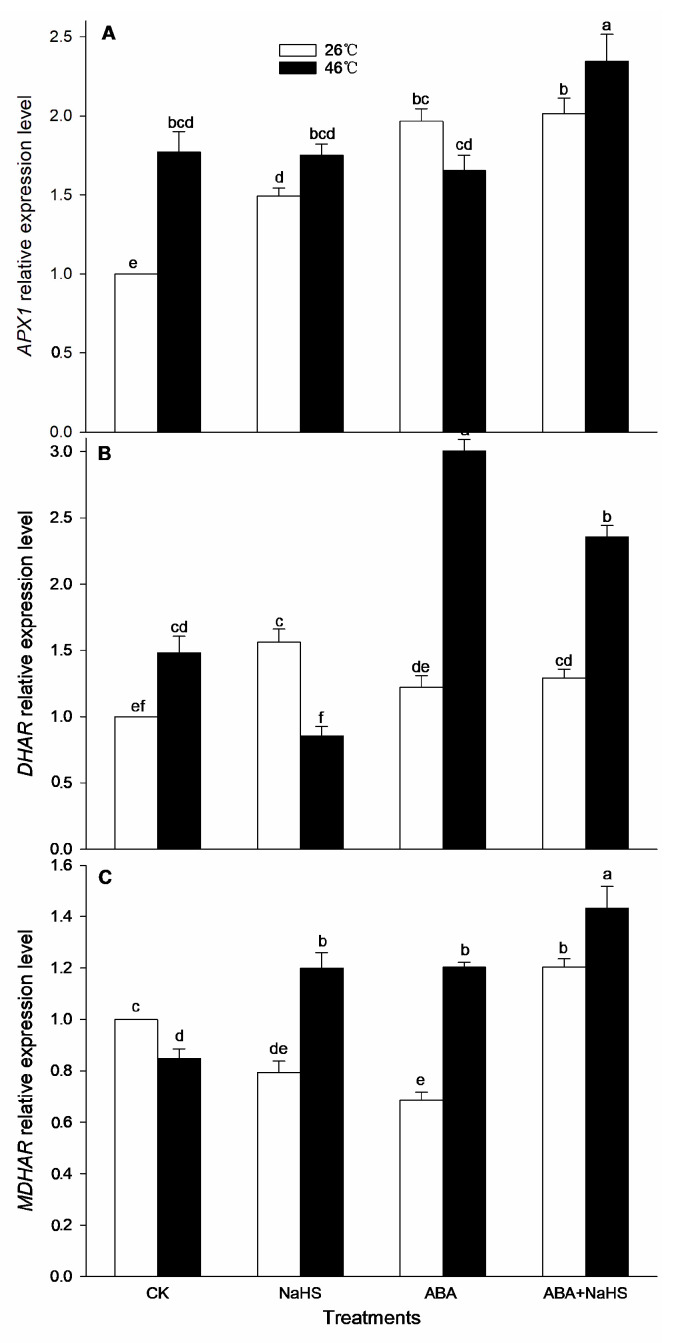
Effect of the irrigation with NaHS and abscisic acid (ABA) alone or in combination on the gene expression of *APX1* (**A**), *DHAR* (**B**), and *MDHAR* (**C**) in maize seedlings under non-heat stress (non-HS) and HS conditions. The significance analysis between the data was performed as a one-way analysis of variance (ANOVA) and Duncan multiple-range test at the 0.05 level. The data in the figures represent mean ± standard error (SE, *n* = 5), the same and different letters indicate the insignificant and significant difference, respectively.

**Figure 14 ijms-24-12264-f014:**
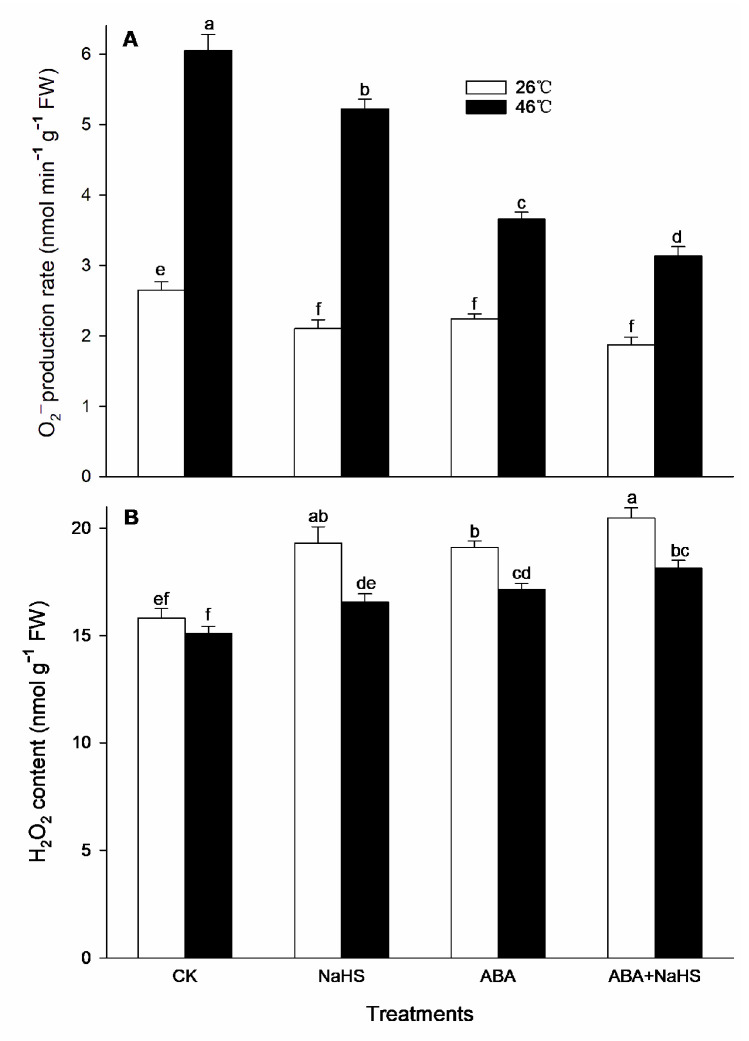
Effect of the irrigation with NaHS and abscisic acid (ABA) alone or in combination on the generation rate for superoxide radical (O_2_^.−^) (**A**) and hydrogen peroxide (H_2_O_2_) content (**B**) in maize seedlings under non-heat stress (non-HS) and HS conditions. The significance analysis between the data was performed as a one-way analysis of variance (ANOVA) and Duncan multiple-range test at the 0.05 level. The data in the figures represent mean ± standard error (SE, *n* = 5), the same and different letters indicate the insignificant and significant difference, respectively.

## Data Availability

All data are displayed in the manuscript and Appendix A.

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
