# Peer review of "The Essential Role of H2S-ABA Crosstalk in Maize Thermotolerance through the ROS-Scavenging System"

_ijms, 2023, doi:10.3390/ijms241512264_

Round 1

Reviewer 1 Report

The manuscript by Wang et al. demonstrates the significance of ABA and H2S crosstalk in thermotolerance through ROS scavenging system. The experiment was well planned and executed to substantially prove ABA and H2S role in thermotolerance using their biosynthesis inhibitors as chemical tool and employing molecular methods. The results show that the combined treatment of ABA and H2S more conspicuously reversed the effects of heat stress in maize. The MS, however, needs a revision to focus on the study and description of the results. Authors are suggested to pay attention to the following recommended changes.

- please consider change of the title to: The essential role of H2S-ABA crosstalk in thermotolerance through ROS-scavenging system

- In the first two para of Introduction, a long description of H2S and ABA biosynthesis is included. This should be reduced to carry only meaningful part of H2S and ABA synthesis in plants. Moreover, the last para should provide details (more studies) on the involvement of H2S and ABA individually or in combination in abiotic stress, particularly thermotolerance.

- Results should include the per cent changes observed by the individual and combined treatment application in the studied parameters compared to control. This should be done throughout the result section to emphasize the changes made by the individual and combined application of H2S and ABA.

- Discussion is well written, but the interaction of H2S/ABA with other phytohormones in thermotolerance should be included to provide support and strength to the discussion. Following published literature will help in better organizing the discussion.

Antioxidants 11: 1478 (2022) doi.org/10.3390/antiox11081478.

Antioxidants 11: 372 (2022) doi.org/10.3390/antiox11020372

Plants 10: 1778 (2021) doi.org/10.3390/plants10091778

- Figure legends and tables should provide details of experimentation so that reader can understand the results without consulting text. Authors are suggested to include treatments details, concentration, time of application and time of observation, statistics details, abbreviation expanded, number of biological samples. The Statistical analysis section in the text will be there.  

Minor editing is required.

Author Response

- please consider change of the title to: The essential role of H2S-ABA crosstalk in thermotolerance through ROS-scavenging system

ANSWER: Thanks for your precious comments! We think this is a proper change. The title has been changed to " The essential role of H2S-ABA crosstalk in Maize thermotolerance through ROS-scavenging system ". Please see TITLE!

- In the first two para of Introduction, a long description of H2S and ABA biosynthesis is included. This should be reduced to carry only meaningful part of H2S and ABA synthesis in plants. Moreover, the last para should provide details (more studies) on the involvement of H2S and ABA individually or in combination in abiotic stress, particularly thermotolerance.

ANSWER: Thanks for your precious comments! The first two paragraphs have been rewritten according to the comments, and the last paragraph was supplemented some contents on plant thermotolerance. Please see INTRODUCTION!

- Results should include the per cent changes observed by the individual and combined treatment application in the studied parameters compared to control. This should be done throughout the result section to emphasize the changes made by the individual and combined application of H2S and ABA.

ANSWER: Thanks for your precious comments! The per cent changes observed by the individual and combined treatment application in the studied parameters compared to control were supplemented, and we found that the survival rate was increased to and % from % of the control after treatment with H2S, ABA, and in combination. Please see RESULTS!

- Discussion is well written, but the interaction of H2S/ABA with other phytohormones in thermotolerance should be included to provide support and strength to the discussion. Following published literature will help in better organizing the discussion.

Antioxidants 11: 1478 (2022) doi.org/10.3390/antiox11081478.

Antioxidants 11: 372 (2022) doi.org/10.3390/antiox11020372

Plants 10: 1778 (2021) doi.org/10.3390/plants10091778

ANSWER: Thanks for your precious comments! the interaction of H2S/ABA with other phytohormones in thermotolerance has been supplemented and further supported the fact that the essential role of H2S/ABA crosstalk in maize thermotolerance. Also, H2S/ABA interacted with other hormones and signal molecules and the three references were added and further discussed. Please see DISCUSSION!

- Figure legends and tables should provide details of experimentation so that reader can understand the results without consulting text. Authors are suggested to include treatments details, concentration, time of application and time of observation, statistics details, abbreviation expanded, number of biological samples. The Statistical analysis section in the text will be there.  

ANSWER: Thanks for your precious comments! We think if the detailed experimentation including the concentration, time, and treatments was added to figure legends, which will lead to the more and longer representation. In general, statistical analysis should be added. So the statistical analysis in figure legends and tables have been provided according to your comments, the more detailed should be refer to METHODS. Please see Figure legends and Tables!

Reviewer 2 Report

The manuscript entitled "Essential Role of ROS-Scavenging System in H2S-ABA Crosstalk-Induced Thermotolerance in Maize Seedlings" presents interesting data; in 3/4 parts, the manuscript is well-written. Some critical remarks exist, i.e.,

- no hypothesis;

- in figures: meaning letters should be explained;

- in tables: meaning asterisks should be explained;

- no number of biological replicates: see figures and text in section 4.8;

- many parts of the discussion are not based on known facts - this part should be rewritten.

Author Response

-No hypothesis

ANSWER: Thanks for your precious comments! The hypothesis has been supplemented in INTRODUCTION. Please see INTRODUCTION!

- in figures: meaning letters should be explained;

ANSWER: Thanks for your precious comments! The meaning letters in figures have been explained accordingly. Please see figure legends!

- in tables: meaning asterisks should be explained;

ANSWER: Thanks for your precious comments! The meaning asterisks in tables have been explained accordingly. Please see table titles!

- no number of biological replicates: see figures and text in section 4.8;

ANSWER: Thanks for your precious comments! The number of biological replicates in figure legends and text in section 4.8 has been supplemented accordingly. Please see figure legends and section 4.8!

- many parts of the discussion are not based on known facts - this part should be rewritten.

ANSWER: Thanks for your precious comments! The parts of the discussion are not based on known facts were rewritten accordingly and supplemented some relative references. Please see highlights in DISCUSSION!

Reviewer 3 Report

Dear Authors,

The manuscript entitled: Essential Role of ROS-Scavenging System in H2S-ABA Crosstalk-Induced Thermotolerance in Maize Seedlings deals with an interesting issue related to the reaction of plants to exogenous application of H2S and ABA. The subject of the publication is interesting, but there is no application of these substances. So what could be the practical use of this knowledge, e.g. in agriculture. Please underline this in the Introduction and in the Conclusions. In addition to this remark, here are my suggestions for the manuscript:

- The Astract is a good introduction to the described issue, but I believe that the full names of enzymes should be followed by abbreviations in parentheses

- Introduction - no information why corn was used for research, why it is important. I am asking for an introduction, characteristics of this species and information about the state of cultivation in China

- Results - the chapter is well described, change the size and type of font on the graphs in accordance with the recommendations of the journal

- Discussion - please move the tables with correlations from this chapter to Results, describe them, but the discussion should only refer to the results of research by other authors

- Materials and Methods - Statistical analysis - no statistical program used to perform the statistical analysis, no description of the correlation analysis

- Conclusions - please provide more specific conclusions about the effect of H2S and ABA on maize seedlings and write about the possibility of practical application of these studies.

Author Response

-Please underline this in the Introduction and in the Conclusions. In addition to this remark, here are my suggestions for the manuscript:

ANSWER: Thanks for your precious comments! The contents of practical use of H2S and ABA in agriculture have been underlined in INTRODUCTION and CONCLUSION according to your comments. Please see INTRODUCTION and CONCLUSION!

- The Astract is a good introduction to the described issue, but I believe that the full names of enzymes should be followed by abbreviations in parentheses

ANSWER: Thanks for your precious comments! The full names of enzymes and antioxidants have been followed by abbreviations in parentheses. Please see ABSTRACT!

- Introduction - no information why corn was used for research, why it is important. I am asking for an introduction, characteristics of this species and information about the state of cultivation in China

ANSWER: Thanks for your precious comments! The contents "maize is the third food crop, the variety was widely planted due to its multiple-stress tolerance and high yield in Yunnan China " was supplemented in INTRODUCTION. Please see INTRODUCTION!

- Results - the chapter is well described, change the size and type of font on the graphs in accordance with the recommendations of the journal

ANSWER: Thanks for your precious comments! The size and type of font on the graphs have been changed according to the requirement of the journal. Plant see graphs!

- Discussion - please move the tables with correlations from this chapter to Results, describe them, but the discussion should only refer to the results of research by other authors

ANSWER: Thanks for your precious comments! The tables with correlations were moved to RESULTS and relative contents were described. Also, other research results were discussed accordingly. Please see RESULTS and DISCUSSION!

- Materials and Methods - Statistical analysis - no statistical program used to perform the statistical analysis, no description of the correlation analysis

ANSWER: Thanks for your precious comments! The statistical program used to perform the statistical analysis and description of the correlation analysis have been supplemented in METHODS. Please see METHODS!

- Conclusions - please provide more specific conclusions about the effect of H2S and ABA on maize seedlings and write about the possibility of practical application of these studies.

ANSWER: Thanks for your precious comments! The specific conclusions about the effect of H2S and ABA on maize seedlings and the possibility of practical application have been supplemented. Please see CONCLUSION!

Round 2

Reviewer 1 Report

The manuscript has been improved and fits well for acceptance.

Reviewer 2 Report

The manuscript has been corrected according to my suggestions

Reviewer 3 Report

Dear Authors,

The manuscript has been revised according to my suggestions. I recommend its publication.